# UniGist: Towards General and Hardware-aligned Sequence-level Long Context Compression

**Chenlong Deng**[1†]**, Zhisong Zhang**[3*]**, Kelong Mao**[1]**, Shuaiyi Li**[2]**,**
**Tianqing Fang**[2]**, Hongming Zhang**[2]**, Haitao Mi**[2]**, Dong Yu**[2]**, Zhicheng Dou**[1*]

[1]Renmin University of China  [2]Tencent AI Lab  [3]City University of Hong Kong
{dengchenlong, dou}@ruc.edu.cn, zhisong.zhang@cityu.edu.hk

## Abstract

Large language models are increasingly capable of handling long-context inputs, but the memory overhead of key-value (KV) cache remains a major bottleneck for general-purpose deployment. While various compression strategies have been explored, sequence-level compression, which drops the full KV caches for certain tokens, is particularly challenging as it can lead to the loss of important contextual information. To address this, we introduce UniGist, a sequence-level long-context compression framework that efficiently preserves context information by replacing raw tokens with special compression tokens (gists) in a fine-grained manner. We adopt a chunk-free training strategy and design an efficient kernel with a gist shift trick, enabling optimized GPU training. Our scheme also supports flexible inference by allowing the actual removal of compressed tokens, resulting in real-time memory savings. Experiments across multiple long-context tasks demonstrate that UniGist significantly improves compression quality, with especially strong performance in detail-recalling tasks and long-range dependency modeling.

## 1 Introduction

As large language models (LLMs) are applied to more sophisticated and demanding applications, the ability to process long-range context has emerged as a fundamental requirement [29, 59]. A wide range of real-world applications, such as retrieval-augmented generation and long-form document understanding, require models to retain and reason over input sequences with extensive lengths [16, 60]. This has spurred growing interest in scaling up context windows from a few thousand tokens to hundreds of thousands or even millions, enabling models to incorporate more history, maintain coherence, and ground responses in broader contexts [23, 35, 46, 14].

Despite this progress, long-context modeling still remains a challenge due to the inherent compute and memory bottlenecks of the underlying Transformer architecture [37]. In particular, the cost of self-attention scales quadratically with sequence length, making both training and inference increasingly resource-intensive as the context grows [2, 50]. More critically, key-value (KV) caching, which is essential for efficient autoregressive decoding during inference, has also become a major source of memory consumption [18, 36]. In long-context settings, the memory required to store KV caches can even exceed that of the model's parameters. These limitations highlight the need for effective KV compression techniques that can reduce computational and memory overhead without sacrificing the model's ability to retain and reason over essential contextual information.

This work focuses on *sequence-level compression*, which directly reduces the number of token representations along the sequence dimension [43, 58]. A particularly promising and general sequence-

---

[†]This work was done during internship at Tencent AI Lab.
[*]Corresponding authors.

level strategy is the gist token-based approach [27, 17, 26, 30, 6, 54, 11], which segments long sequences into chunks and inserts learnable virtual tokens (i.e., gists) to represent the original tokens. In principle, this method has potential for dramatically reducing sequence length, as gist tokens can be much fewer than raw tokens.

However, we observe two major drawbacks in existing gist-based methods. First, they suffer from notable degradation when required to recall information from distant chunks. We hypothesize that this is due to the prevalent chunk-wise training scheme, which introduces a shortcut letting tokens to minimize loss by attending only to nearby uncompressed tokens within the same chunk [6, 54]. As a result, the model is less inclined to learn to integrate information across chunk boundaries, limiting its ability to reason over long-range dependencies. Second, the chunk-wise design suffers from fragmented memory usage and complex computational graphs, which hampers overall training efficiency. In this training scheme, chunks are processed in an iterative style and separate memory allocations are required for each chunk, leading to potential under-utilization of hardware resources.

In this work, we address the above challenge by introducing **UniGist**, a unified gist-based framework for context compression. Unlike prior approaches that rely on chunk-wise training, we remove this constraint and enable the model to learn compression more effectively over the entire sequence through a unified sparse gist layout. To further enhance the efficiency of this layout, we propose a gist shift trick, which is a hardware-aligned kernel design that transforms the irregular, vertically sparse attention pattern into a right-aligned block structure. This transformation aligns with the block-wise execution patterns of modern GPU architectures, significantly enhancing training throughput as well as supporting fast autoregressive inference. Experiments over a wide range of long-context tasks illustrate the effeceveness and efficiency of the proposed approach.

Our main contributions are as follows:

1) We propose UniGist, a sequence-level compression method with a unified sparse gist layout, enabling effective and efficient long context modeling without chunk-wise training.

2) We introduce the gist shift trick to align the proposed sparse attention layout with GPU-friendly block structures, significantly improving training throughput and supporting efficient inference.

3) With experiments on a variety of long-context tasks, we demonstrate that UniGist can provide improvements for context compression, with especially strong performance in detail-recalling tasks and long-range dependency modeling.

## 2  Related Work

**KV Cache Compression.**    Transformer-based large language models mostly employ key-value caching to avoid repetitive attention computations during inference. However, as real-world applications demand ever-increasing context lengths, the associated memory and bandwidth overhead has quickly become a primary bottleneck for real-time systems. To tackle this, KV compression techniques are widely explored along four orthogonal dimensions (i.e, layer, head, token sequence, and numerical precision). *Layer-level* strategies typically skip certain transformer layers or share activations across layers [33, 34, 3, 32]. At the *head level*, multiple query heads can share a single key head, or the long-term cache of specific heads can be eliminated [1, 31]. Recent low-rank approaches further introduce a promising direction for this line [10, 56]. *Precision-focused* methods apply custom quantization and control the floating-point error they introduce to maintain the model's ability [24]. Moreover, some techniques within the layer, head, and precision dimensions demonstrate near-lossless performance on general long context tasks, making them broadly applicable in cutting-edge models.

On the other hand, *sequence-level* compression falls into two broad categories. **Token eviction** discards older token activations in real time once a budget is exceeded. For example, StreamingLLM [43] retains only the initial and recent tokens, which irreversibly lose the details in evicted tokens. To mitigate this, most of the following methods [58, 22, 47, 4, 12, 13] have to rely on hybrid eviction of layer or head level, or even resort to question heuristics rather than general compression. **Token merging** aggregates multiple tokens to reduce the overall cache size [57, 38, 40]. Models that introduce new "gist" tokens and adopt more training often achieve more stable compression [27, 6, 54]. Nonetheless, prior study shows that gist-based compression still struggles to retain highly complex details [11].

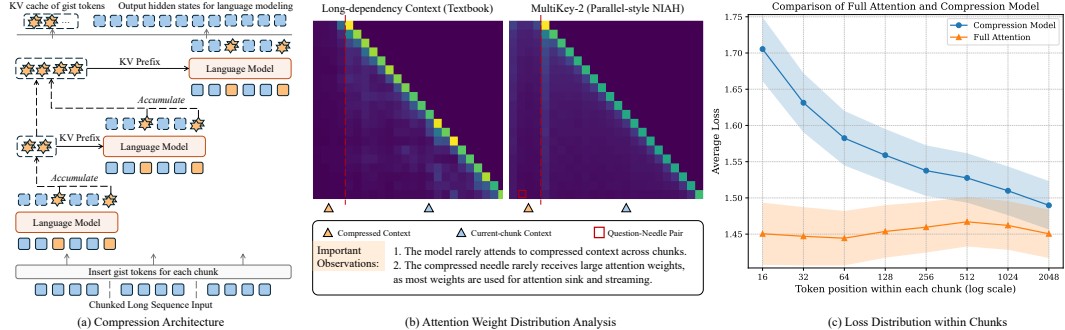

Figure 1: Analysis of previous architectures. (a) Previous approaches adopt a chunk-wise strategy, which accumulates gist representations iteratively. (b) Our attention analysis reveals that tokens tend to ignore compressed contexts outside the current chunk. (c) Our token-position analysis indicates that tokens near the chunk ends may rely only on uncompressed tokens to minimize the target loss.

Our method addresses the key limitations of gist-based architectures, achieving high-quality and general-purpose compression using sequence-level alone.

**Sparse Attention.** Sparse attention has been widely adopted to mitigate the quadratic complexity of self-attention by restricting computation to a subset of token interactions. Empirical studies show that attention maps in language models are inherently sparse, enabling full attention can be approximated closely. Early methods focus on designing fixed sparsity patterns and training models to adapt to them [2, 51, 50]. With large language models scaling, researchers find that models trained with full attention could preserve performance by simply adopting dynamically searched sparsity patterns without extra tuning [20, 42, 41, 44, 53]. More recent works train dynamic attention patterns from scratch [15, 25, 49], in some cases even outperforming full attention. However, to ensure generality across diverse inputs, state-of-the-art sparse attention methods still require the cache for each token to maintain reachability. Our method leverages the intrinsic sparsity of the gist token–based architecture, eliminating the need for distant original token caches and achieving substantial acceleration during both training and inference.

## 3 Why Does Typical Gist-based Compression Fail in Details?

### 3.1 Preliminary

Previous empirical study [11] shows that the most effective variant of gist token-based compression inserts learnable gist tokens in an interleaved manner within each fixed-length chunk, and retains the KV cache of gist tokens across all transformer layers. This section introduces this variant as the representative architecture for our analysis.

Figure 1(a) illustrates the overall architecture. Given a long sequence $X = [x_1, x_2, \ldots, x_T]$, a chunk length $L$, and a compression ratio $r$, the sequence is first divided into $N = T/L$ fixed-length chunks. Within each chunk, one gist token is inserted after every $r$ raw tokens to compress the sequence. As a result, each chunk contains $L$ raw tokens and $n = L/r$ interleaved gist tokens.[1] The $i$-th chunk's content is given by:

$$\tilde{X}^{(i)} = [\underbrace{x_1^{(i)}, \ldots, x_r^{(i)}, g_1}_{\text{the 1-st gist unit}}, \ldots, \underbrace{x_{L-r+1}^{(i)}, \ldots, x_L^{(i)}, g_n}_{\text{the } n\text{-th gist unit}}], \quad r < L, \tag{1}$$

where $g$ denotes the inserted gist tokens. Each group consisting of $r$ raw tokens and the subsequent gist token is defined as a **gist unit**. For simplicity, all inserted gist tokens share the same embedding. For the $i$-th chunk, its input to the transformer blocks contains not only the current $\tilde{X}^{(i)}$, but also the accumulated KV cache from all preceding chunks' gist tokens $G_{<i}$ as the prefix, thereby enabling information flow across chunks. To maintain consistency with the inference stage, the same chunking

---

[1]Here we assumes that $T$ can be divided by $L$ and $r$ for clarity.

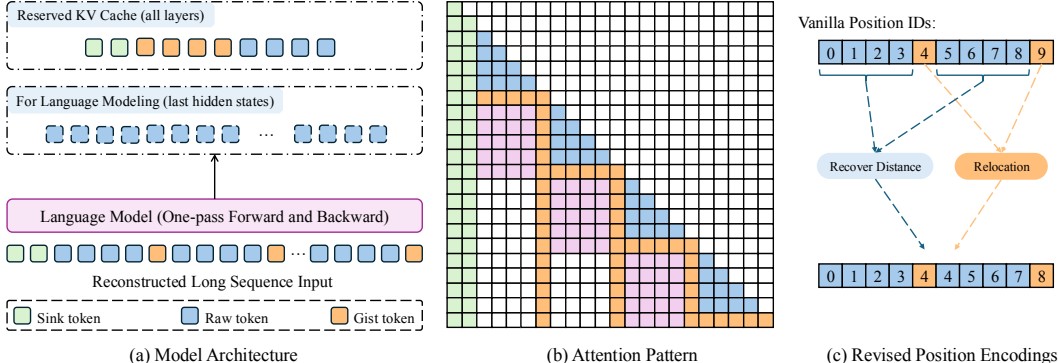

| (a) Model Architecture | (b) Attention Pattern | (c) Revised Position Encodings |

Figure 2: Overall architecture of UniGist. (a) We adopt a chunk-free training strategy that allows one-pass sequence processing. (b) We design a gist-enhanced attention pattern to effectively aggregate information. (c) We revise positional encoding to maintain the relevant distances between raw tokens.

and iterative chunk-size forward strategy are used during training. The final training objective follows the standard cross-entropy language modeling loss over raw tokens.

## 3.2 Analysis

We analyze why previous gist-based models struggle to fully utilize compressed context from two perspectives by experimenting on a well-trained gist-based model:

**Attention Patterns.** We examine the attention patterns on two datasets with long-range dependencies (Figure 1(b)). In natural data such as textbooks, most tokens exhibit little attention across chunk boundaries. For structured tasks like RULER MultiKey-2 [19], which follow a parallel needle-in-a-haystack pattern, attention also remains mostly localized within the current chunk. In addition, many attention weights concentrate at the beginning of each chunk, even though this region may contain irrelevant key-value pairs. In contrast, the actual related piece of context receives little attention.

**Token Positions.** We perform continue-training with a well-trained compressed model on 32K-length sequences and measure the average loss at each relative position within chunks, excluding the first chunk that has no previous contexts. As shown in Figure 1(c), tokens at the beginning of each chunk consistently exhibit higher loss, while those closer to the end show loss patterns similar to full attention models. This suggests that tokens near the end can rely on nearby uncompressed tokens to minimize the language modeling loss. As a result, previous contexts may not be effectively used for compression learning, which limits the overall efficiency of training.

These observations suggest that chunk-wise training is a primary bottleneck limiting compression quality. Previous approaches rely on chunk-wise training to ensure consistency between training and inference, but this strategy prevents models from learning cross-chunk dependencies and limits compression performance, calling for better designs for training strategies.

## 4 Method

### 4.1 Unified Gist Context-based Language Modeling

As illustrated in Section 3.2, the chunk-wise training leads to an under-attention problem, that is, tokens in each block tend to ignore earlier compressed contexts. To address this, we eliminate the coarse-grained chunking strategy and introduce a unified attention pattern to enable the language model to learn more effective context compression. Figure 2 provides an overview of our strategy. Given a token sequence $[x_1, x_2, \ldots, x_T]$, we construct the augmented input with the following steps:

1) **Attention Sink Prefix.** We prepend a fixed number of $s$ sink tokens $[s_1, s_2, \ldots, s_s]$ at the beginning of the entire sequence. They serve as fixed "attention sinks" [43] to prevent mode collapse that can occur when initial raw tokens are compressed and then removed.

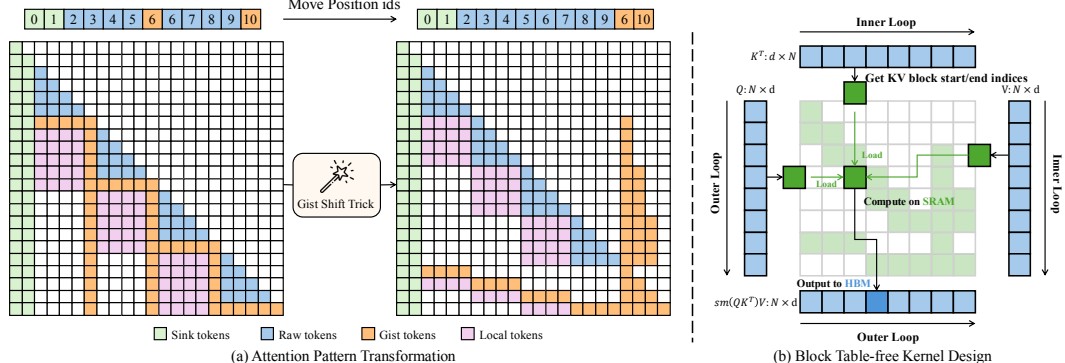

Figure 3: Kernel design of our efficient sparse gist attention. (a) We adopt the gist shift trick by moving all gist tokens to the rightmost position of the sequence to facilitate GPU block processing. (b) With fixed compression rate, non-visible blocks can be skipped by pre-calculating block indexes.

2) **Gist Token Insertion.** We interleave learnable gist tokens into the token sequence with a compression ratio $r$ (i.e., one gist token for every $r$ raw tokens). We uniformly apply this insertion operation over all raw tokens. After this operation, the new sequence will be:

$$Z = [s_1, \ldots, s_s, x_1, \ldots, x_r, g_1, x_{r+1}, \ldots, x_{2r}, g_2, \ldots, x_T, g_{T/r}]$$
$$= [z_1, \ldots, z_{T'}], \quad T' = s + T + T/r \tag{2}$$

3) **Position Encoding Alignment.** Inserting gist tokens shifts the position indices of later tokens, which changes the original distances between raw tokens and may harm language coherence. To mitigate this, we assign each gist token the same position id as the raw token right after it, so that all raw-token distances remain unchanged.

With this gist-augmented input, we define a unified sparse autoregressive attention pattern that is applied to *all tokens*. For each $z_t \in Z$, the set of its visible tokens is defined as:

$$\mathcal{A}(z_t) = (\mathcal{S} \cup G \cup \mathcal{W}_t) \cap \{z_j | j \le t\}, \tag{3}$$

where the intersection with $\{z_j | j \le t\}$ enforces the autoregressive constraint, restricting each token to attend only to itself or prior tokens. $S$ and $G$ are sink and gist tokens, respectively, which can provide conpressed contexts. The local window $W_t$ consists of the current gist unit that contains $z_t$, as well as the previous $k$ gist units (each containing $r$ raw tokens followed by one gist token). This helps the model retain its basic language modeling ability with local contexts. The final training objective remains the standard autoregressive cross-entropy loss over raw tokens $x_t \in X \subset Z$.

## 4.2 Hardware-aligned Kernel Design

Sparse attention patterns in gist-token-based models are inherently efficient. Nevertheless, to fully realize such efficiency on GPU devices, we need suitable hardware-aligned kernels. Because of the sparsely inserted gist tokens, our attention patterns are incompatibility with FLASH_ATTN kernels [8]. Therefore, we design a custom kernel that supports efficient processing.

Since our attention pattern evenly distributes the gist tokens in a fine-grained way, conventional GPU kernels cannot fully exploit the advantages of sparse attention. The main reason is that GPU kernels are optimized for processing data in blocks (typically of size 64 or 128). In our setting (with a compression ratio of 4 or 8), gist tokens are always attended to and are present in every block. As a result, no blocks can be skipped during attention computation in training, even though the raw tokens in previous blocks outside the local window are not visible to later tokens and can be skipped.

We address this issue with the **gist shift trick**, which moves all gist tokens to the rightmost position of the sequence. This transformation converts the attention pattern into a dense, right-aligned scheme that aligns well with standard GPU block processing, as shown in Figure 3. In this way, all the gist tokens are gathered together, and each token will only need to attend to the compacted gist blocks in addition to the sink and local window blocks. This leads to a scheme that can fully exploit the sparsity benefits by skipping non-attended blocks. Note that with a fixed compression ratio, our attention

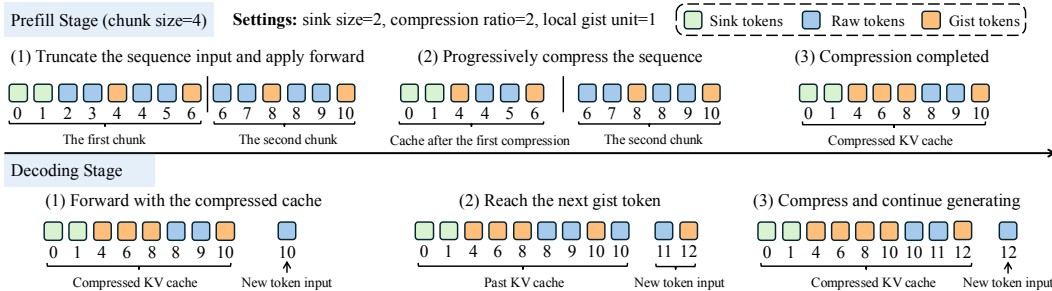

Figure 4: Illustration of flexible inference. Since the raw tokens outside the local window will never be attended to by future tokens, these invisible raw tokens can be safely discarded at each compression point. For prefilling, we can re-adopt the chunk-wise scheme and encode each chunk of the input with our efficent kernel, with nonvisible tokens dropped after each chunk. For decoding, nonvisible raw tokens can be discarded after each new gist is inserted.

layout is deterministic, and there is no need to build or store any block index tables. Instead, the indexes of the visible blocks for each token can be directly pre-calculated.

This trick brings a side effect that the original token indexes are distorted, which leads to the difficulty of calculation **in-block attention masks**. To mitigate this issue, we pass an auxiliary input of the original token indexes to the kernel. With this information, we can easily calculate the attention masks according to Equation 3.

## 4.3  One-pass Training and Flexible Inference

**Training: One-pass with Full Sequence.**   With our unified attention pattern and custom kernel, we can process long sequences in a single pass without chunk-wise iterative processing during training. This eliminates memory fragmentation and reduces the complexity associated with the chunk-based scheme, leading to faster and more memory-efficient training.

**Inference: Flexible Decoding with Low Memory Usage.**   For prefilling, which naturally follows the iterative manner, we can re-adopt the chunk-wise scheme to encode each chunk with our kernel to reduce peak memory usage.[2] The overall procedure is illustrated in Figure 4. During inference, each token only attends to tokens defined by the autoregressive pattern. Raw tokens beyond the local window are no longer required and can be dropped on the fly during generation. This also allows compatibility with efficient decoding techniques like FlashDecoding [9].

## 5  Experiments

### 5.1  Experimental Setup

**Baselines.**   We compare UniGist primarily with methods that perform sequence-level context compression, as compression techniques along other dimensions are often orthogonal and not directly comparable. Our baselines include three categories of methods: (1) Vanilla Full Attention Models, including the original model and its supervised fine-tuned variant using the same data as ours to control for variables. (2) Training-free Context Compression, where we include StreamingLLM [43], SnapKV [22], PyramidKV [4], and AdaKV [12] (based on SnapKV) as popular eviction methods. (3) Gist Token-based Compression, where we evaluate AutoCompressors [6] and Activation Beacon [54] as state-of-the-art methods. Both use chunk-wise compression to enable general-purpose compression.

**Implementation Details.**   We use Llama3.1-8B-Instruct and Llama-3.2-3B-Instruct as the base models to evaluate performance across different model sizes [35]. Continued pretraining is conducted on 16B tokens of 32K-length samples drawn from Prolong's [14] mixed dataset, followed by supervised tuning on 1B tokens of mixed instruction data. Cross-document masking is applied during training to block attention across document boundaries. All custom attention kernels are implemented in Triton. For all methods, we adopt a question-agnostic compression setup to ensure generality and

---

[2]The chunk size can be configured as any integer multiple of the compression ratio during prefilling.

| Method | RAG | Rerank | LongQA | ICL | Synthetic | Summ. | Average |
|---|---|---|---|---|---|---|---|
| *Llama-3.1-8B-Instruct* | | | | | | | |
| Full Attention | 74.5 | 55.1 | 43.5 | 81.8 | 99.3 | 28.9 | 63.9 |
| Full Attention (FT) | 72.7 | 48.4 | 43.8 | 79.5 | 97.4 | 26.8 | 61.4 |
| StreamingLLM | 58.5 | 28.4 | 33.1 | 36.0 | 18.6 | 12.2 | 31.1 |
| SnapKV | 60.3 | 11.8 | 40.4 | 29.6 | 19.8 | 12.6 | 29.1 |
| PyramidKV | 60.9 | 9.3 | 41.3 | 22.4 | 21.7 | 12.7 | 28.0 |
| AdaKV | 61.1 | 14.0 | 40.5 | 37.6 | 31.6 | 12.5 | 32.9 |
| AutoCompressors | 64.7 | 20.9 | 36.4 | 40.1 | 23.2 | 16.5 | 33.6 |
| Activation Beacon | 67.9 | 34.4 | 40.6 | 79.2 | 61.8 | 21.0 | 51.8 |
| UniGist (Ours) | **71.3** | **45.5** | **45.5** | **85.8** | **91.3** | **22.9** | **60.4** |
| *Llama-3.2-3B-Instruct* | | | | | | | |
| Full Attention | 68.6 | 17.5 | 38.6 | 79.6 | 78.9 | 28.2 | 51.9 |
| Full Attention (FT) | 66.3 | 16.9 | 39.1 | 77.1 | 85.6 | 27.7 | 52.1 |
| StreamingLLM | 52.3 | 10.9 | 13.2 | 5.8 | 15.1 | 11.5 | 18.1 |
| SnapKV | 52.8 | 3.0 | 29.4 | 8.6 | 16.2 | 12.3 | 20.4 |
| PyramidKV | 52.5 | 3.2 | 31.5 | 7.8 | 12.6 | 11.6 | 20.0 |
| AdaKV | 53.8 | 3.1 | 30.5 | 6.6 | 28.6 | 12.4 | 22.5 |
| AutoCompressors | 55.7 | 7.2 | 26.3 | 31.2 | 7.9 | 15.2 | 23.9 |
| Activation Beacon | 58.4 | 11.4 | 36.5 | 57.4 | 47.1 | 16.8 | 37.9 |
| UniGist (Ours) | **63.6** | **15.6** | **41.7** | **73.8** | **82.3** | **21.6** | **47.8** |

Table 1: Evaluation results of various context compression methods under two base models across long-context tasks. "FT" denotes to the fine-tuned full attention model. Bolded numbers indicate the best results among compression methods.

difficulty. Greedy decoding is used for all tasks. For UniGist, the sink size is set to 128, and the local window corresponds to 128 raw tokens. The main results reported use a compression ratio of 4. All training and inference experiments are conducted using the Huggingface framework. More details about data composition and training setup are introduced in Appendix A.

## 5.2 Main Results

**Long Context Evaluation.** We evaluate models' long-context understanding using the HELMET benchmark [48], which includes a broad range of long-context tasks, including retrieval-augmented generation (RAG), summarization, and synthetic recall. HELMET incorporates several representative datasets such as Inf-Bench [55] and RULER [19], offering a comprehensive assessment of model performance

| Method | MMLU-Pro | GSM8K | HellaSwag |
|---|---|---|---|
| Full | 47.7 | 83.6 | 60.2 |
| Full-FT | 47.1 | 84.3 | 59.8 |
| Beacon | 47.3 | 84.2 | 60.1 |
| UniGist | 47.6 | 83.9 | 60.1 |

Table 2: Performance on three widely-used short-context datasets. The "Full" and "FT" denote to full attention and fine-tuning, respectively.

across diverse long-context scenarios. For each task, we report the average score over its datasets. Detailed task configurations are provided in Appendix B.

Table 1 presents the overall evaluation results. We highlight three key observations: (1) **UniGist achieves the best performance among all compression methods across both model sizes.** On the synthetic tasks such as RULER MK-3, it is the only method that performs close to full attention. (2) **UniGist shows clear improvements over prior methods with similar architectures** (e.g., Beacon). This suggests that our unified attention pattern design plays a key role in helping the model read compressed context effectively. (3) **The effect of compression varies by task.** In tasks like RAG, many methods still perform reasonably well. In contrast, many-shot ICL is much more sensitive, especially for training-free approaches. This may be because compression disrupts the parallel structure required for in-context learning. Gist token-based methods show better robustness under these conditions.

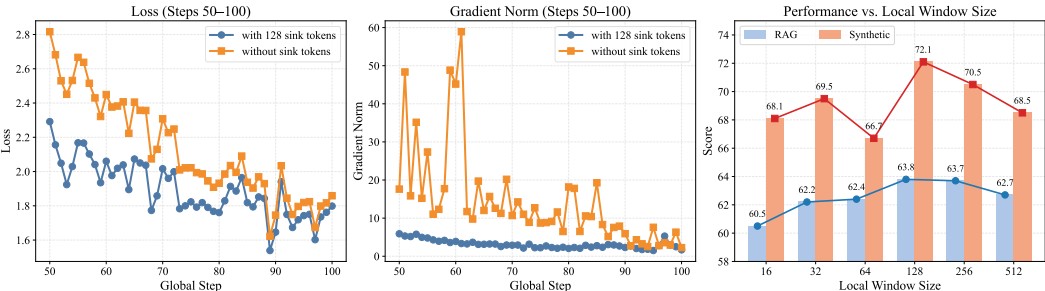

Figure 6: **Left & Middle:** Loss and gradient norm curves of Llama-3.1-8B-Instruct during training steps 50–100. Using 128 sink tokens leads to more stable training, with lower loss and smaller gradient fluctuations. **Right:** Performance of two long-context tasks of Llama-3.2-3B-Instruct after 4B-token continual pretraining and then supervised finetuning under different local window sizes.

**Basic Short-context Capability.** To assess whether context compression compromises the model's core capabilities, we further evaluate all methods on a set of standard short-context benchmarks, including MMLU-Pro [39] (knowledge and reasoning), GSM8K [7] (math), and HellaSwag [52] (commonsense inference) with 3-shot demonstration, as shown in Table 2. These tasks do not involve long contexts and thus serve as a proxy to measure whether compression-specific training introduces undesirable side effects on general performance. We find that all gist token-based methods maintain performance comparable to the full attention baseline across all tasks. Differences between methods are within the range of natural variance, and no consistent degradation is observed. This suggests that continued pretraining with the compression target does not impair the model's ability to perform basic tasks. The gains in long-context understanding come without sacrificing short-context capabilities.

**Boundary Effect Test.** As shown in Section 3.2, chunk-based training tends to introduce boundary effects, where a token's performance depends on its position within the chunk. We evaluate UniGist under the same setting and compare it with full attention. Figure 5 shows that UniGist maintains nearly uniform perplexity across positions, closely matching full attention. This confirms that UniGist avoids position-related bias caused by the chunk-wise training scheme.

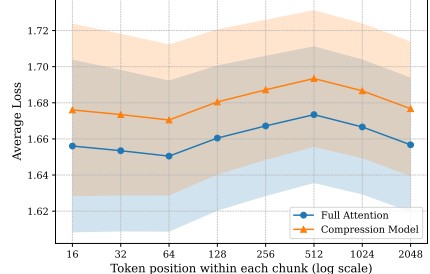

Figure 5: Boundary Effect Test. The compression model exhibits a similar trend to full attention, showing no degradation near chunk boundaries.

### 5.3 Ablation Study

The core mechanism of UniGist is the unified attention layout, which is designed for learning better compression. Its effectiveness is already evident in the main results through the performance comparison with Beacon's chunk-wise training. In this section, we focus on two additional components that contribute to UniGist's performance: the use of sink tokens and the choice of local window size.

**Sink tokens for Stable Learning.** To understand their impact, we compare training behaviors between models with and without sink tokens during the first 50 to 100 training steps. As shown in the left of Figure 6, the model with sink tokens achieves consistently lower loss and smoother gradient norms. In contrast, the model without sink tokens shows frequent gradient spikes, which slow down convergence. One possible reason is that, without sink tokens, the model fails to consistently attend to the initial tokens in the sequence because of progressive compression. This may break the autoregressive pattern and weaken early-stage learning signals.

**Effect of Local Window Size.** We further examine how the size of the local attention window affects the trade-off between language modeling and compression learning. A small window limits the model's access to raw tokens, forcing it to rely more heavily on gist tokens. This encourages the model to learn to extract compressed representations, but also increases the difficulty of language modeling due to reduced contextual continuity. In contrast, a large window gives the model easy

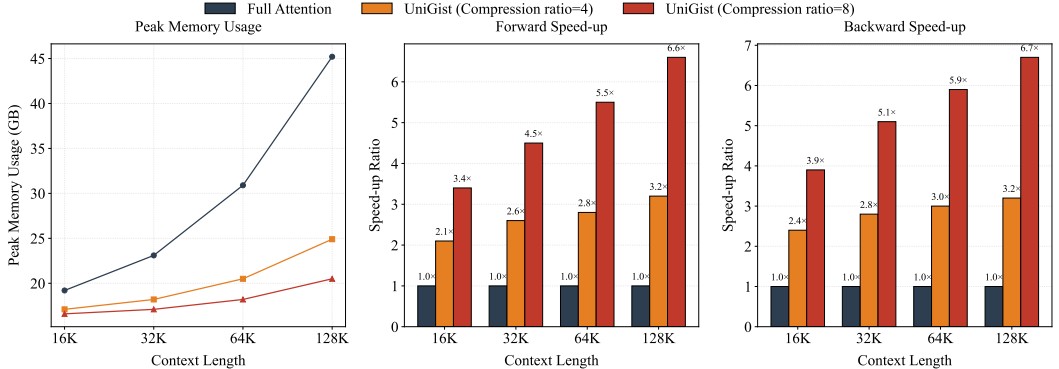

Figure 7: Comparison of efficiency between UniGist and standard full attention. **Left:** Peak memory usage increases steeply with context length for full attention, while UniGist maintains significantly lower memory consumption. **Middle & Right:** UniGist delivers substantial forward and backward attention speed-up, with higher gains under longer sequences and larger compression ratios.

access to raw tokens, which weakens the pressure to use gist tokens and may cause the model to bypass the compression path altogether. To quantify this trade-off, we evaluate UniGist with different window sizes on two long context tasks. As shown on the right of Figure 6, performance on both the RAG and synthetic recall tasks exhibits a non-monotonic pattern, initially improving and then declining. Notably, both tasks achieve peak performance when the window size is set to 128. This suggests that a moderate window provides sufficient local context for stable language modeling, while still encouraging the model to rely on gist tokens for global understanding.

### 5.4 Efficiency Comparison

UniGist is primarily designed to reduce the memory occupation of the KV cache, while also providing substantial speed-ups during both training and inference. We evaluate the quantitative benefits from two perspectives: memory usage and computation speed.

**Memory Usage.** Long-context modeling places heavy demands on memory resources, often becoming a bottleneck for deployment. We evaluate peak GPU memory usage under compression ratios of 4 and 8, comparing UniGist with full attention on the Llama-3.1-8B-Instruct model. As shown in the left of Figure 7, UniGist achieves a substantial reduction in memory usage in both settings, with more pronounced savings under ratio=8. These results demonstrate that UniGist can effectively alleviate memory bottlenecks while preserving modeling performance. The benefits grow with longer contexts, making UniGist a strong choice for scaling to ultra-long sequences.

**Speed-up Evaluation.** To isolate the speed-up benefit of the attention mechanism itself, we measure the forward and backward pass latency of a single attention layer. This avoids confounding factors from other components of the model. As shown in the right of Figure 7, the speed-up achieved by UniGist grows with context length and gradually approaches the theoretical upper bound, which equals the compression ratio (e.g., $4\times$ speed-up under compression ratio=4)[3]. This confirms that the sparse attention path introduced by UniGist leads to a real acceleration in real scenarios, especially in long sequences where the quadratic cost of full attention becomes prohibitive.

## 6 Conclusion

This work presents UniGist, a sequence-level compression framework for long-context language modeling without relying on hybrid strategies. By removing chunk-wise training and introducing a unified gist attention layout, UniGist enables effective learning over compressed sequences. We design a gist shift trick and a block-table-free sparse attention kernel that improve training efficiency and are fully reusable during inference. UniGist also supports flexible chunking at inference time for adaptive context processing. Experiments across a broad range of tasks demonstrate that UniGist

---

[3]We provide the original data and experimental details in Appendix C.

achieves solid performance across a range of tasks, and proves particularly effective at retaining details in long context scenarios. It also improves efficiency over full attention, with faster runtime and lower memory usage during inference.

## 7 Acknowledgments

This work was supported by Beijing Municipal Science and Technology Project No. Z231100010323009, National Natural Science Foundation of China No. 62272467, Beijing Natural Science Foundation No. L233008. The work was partially done at the Engineering Research Center of Next-Generation Intelligent Search and Recommendation, MOE.

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

| Category | Tasks | Metrics |
|---|---|---|
| RAG | NQ | SubEM |
| | TriviaQA | SubEM |
| | PopQA | SubEM |
| | HotpotQA | SumEM |
| Rerank | MS Marco | NDCG@5 |
| Long-doc QA | ∞Bench QA | ROUGE Recall |
| | ∞Bench MC | Accuracy |
| Many-shot ICL | TREC Coarse | Accuracy |
| | TREC Fine | Accuracy |
| | NLU | Accuracy |
| | BANKING77 | Accuracy |
| | CLINIC150 | Accuracy |
| Synthetic recall | JSON KV | SubEM |
| | RULER MK Needle | SubEM |
| | RULER MK UUID | SubEM |
| | RULER MV | SubEM |
| Summ. | ∞Bench Sum | ROUGE-Sum F1 |
| | Multi-LexSum | ROUGE-Sum F1 |

Table 3: Data Composition and metrics of the used HELMET benchmark.

# A  Model and Training Details

## A.1  Baseline Configuration

For training-free compression methods, we adopt the implementation based on the KVPress framework [28] to ensure consistency. For chunk-wise training methods, we divide sequences into 2K-length chunks and continue pretraining and fine-tuning on 32K-length samples. To ensure fairness, all trainable methods are fine-tuned with full-parameter updates. As for vanilla LLaMA models, given their extensive pretraining on approximately 15T tokens, we skip additional pretraining and apply supervised fine-tuning directly.

## A.2  Hyper-parameters

For continued pretraining, we use a batch size of 2M tokens and set the learning rate to 1e-5. The learning rate is warmed up linearly from 0 over 256 steps and then decayed to 50% of its peak using cosine scheduling. The AdamW optimizer is used for our experiments. The compression ratio and local window size remain fixed throughout training. During fine-tuning, we retain most hyperparameters but reduce the warm-up steps to 128.

## A.3  Training Data Processing

For continued pretraining, we use the 64K-length dataset from Prolong [14] and restructure it into 32K-length samples. Specifically, we split all 64K-length samples into two 32K segments. The original data includes two types of samples: those consisting of a single long document, and those formed by concatenating multiple shorter documents. For single-document samples, we apply no additional processing beyond the split. For multi-document samples, we pad each document individually so that its length is divisible by the compression ratio, ensuring compatibility with our attention pattern.

For supervised fine-tuning, we use the Magpie-Llama-3.1-Pro-MT-300K-Filtered dataset [45] as the main source. Then, we refer to the Beacon's [54] setup to further augment it with samples from LongAlpaca [5], BookSum [21], and 500 RULER-like synthetic data designed for long-context tasks. In total, the fine-tuning data comprises approximately 1B tokens.

| Direction | Method | Ratio | 16K | 32K | 64K | 128K |
|-----------|--------|-------|------|-------|-------|--------|
| | Causal | - | 13.8 | 52.7 | 206.9 | 863.9 |
| Forward | UniGist | 4 | 6.45 | 20.5 | 72.9 | 271.8 |
| | UniGist | 8 | 4.10 | 11.6 | 37.3 | 130.9 |
| | Causal | - | 57.4 | 225.6 | 871.5 | 3586.8 |
| Backward | UniGist | 4 | 24.3 | 80.9 | 294.7 | 1116.8 |
| | UniGist | 8 | 14.9 | 44.5 | 147.9 | 538.9 |

Table 4: Forward and backward time (ms) under different context lengths. UniGist is evaluated at compression ratios (CR) of 4 and 8.

## B  Evaluation Details

**Long Context Evaluation.**   We evaluate on six tasks from the HELMET benchmark. The datasets and metrics are listed in Table 3. We use a chat template for all prompts. For the LongQA task, we use a 2-shot demonstration to ensure an exact match in multi-choice questions. Since most models are trained with a 32K context length, we use the corresponding 32K configuration files. For tasks like Rerank, we append a brief reminder after the question to reinforce the expected output format and task objective.

## C  More Experiments

**Speed-up Details**   We report the raw measurements of speed-up under a controlled setting with batch size 1, 32 attention heads, and a head dimension of 128. All methods are implemented with Triton to ensure a fair comparison. Each result is averaged over five runs, as shown in Table 4.

## D  Limitations

**Model Scale and Context Length**   Our experiments are limited to models with at most 8B parameters and a context length of 32K due to computational constraints. Although this setup enables controlled and efficient experiments, it does not fully capture the potential of larger models to learn and benefit from compression. Understanding how model scale and context length influence compression quality remains an open question for future research.

**Training Protocol**   In this work, we adapt existing language models to our attention pattern through continued pretraining. While we do not train from scratch, the UniGist architecture is fully compatible with end-to-end training. Training from scratch may allow the model to better internalize the structure and inductive biases of our compression pattern, without being influenced by pre-existing attention mechanisms. However, such training typically requires hundreds of billions of tokens to yield meaningful insights, which is beyond our current computational budget. We leave a comprehensive investigation of scratch training for future work, especially to understand how compression-aware architectures behave during early-stage pertaining.

## E  Ethical Discussion

UniGist aims to reduce the computational and memory overhead of large language models by introducing efficient context compression. This brings tangible benefits: it lowers the cost of inference, reduces carbon footprint, and facilitates the deployment of long-context models in resource-constrained environments. However, compression may inevitably alters model behavior. Compared to full attention baselines, compressed models may yield less accurate or misleading outputs in certain edge cases, especially when critical information is omitted or distorted during compression. This poses a risk in high-stakes applications where incorrect responses can have real-world consequences. We urge users to carefully evaluate reliability when applying UniGist in downstream tasks and to consider human oversight when appropriate.

