# OpenReview forum: "UniGist: Towards General and Hardware-aligned Sequence-level Long Context Compression"
_NeurIPS.cc/2025/Conference — NeurIPS 2025 poster_

### Official Review · Reviewer_7dFr · 2025-06-21

**Clarity:** 3
**Significance:** 2
**Originality:** 2
**Rating:** 4
**Confidence:** 3

**Summary:**

To address the accuracy challenge of the gist token-based KV cache compression approach, this work proposes a unified gist-based framework for effective compression, named UniGist. By discarding the chunk-wise compression scheme, the LLMs can learn the compression more effectively over the entire sequence. Extensive experiments validate that UniGist outperforms the existing gist token-based compression methods in terms of algorithm accuracy. By implementing the GPU kernel specifically, the proposed method also achieves higher inference throughput.

**Questions:**

See above.

**Ethical Concerns:**

["NO or VERY MINOR ethics concerns only"]

**Final Justification:**

After reading the rebuttal, I think the author has adequately addressed my main concerns. Specifically, the author further clarifies the superiority and flexibility of the proposed UniGist compared to conventional chunk-wise compression methods. The supported maximum context length on a Single GPU was also evaluated and supplemented. I believe the main contributions are solid and reasonably supported, and the paper may be of interest to the community. Thus, I am raising my rating to 'Borderline Accept'.

**Limitations:**

Yes

**Quality:**

2

**Strengths And Weaknesses:**

[Strengths]:
1. The motivation of the proposed method is demonstrated clearly, and the paper is well-written.
2. The GPU kernel for UniGist is implemented and achieves actual throughput improvement.


[Weaknesses]:
1. The proposed scheme lacks logical rationality. Unlike the full sequence processing, the chunk-wise processing is proposed to satisfy the memory constraint throughout the full inference process in the case of long text. Adopting full sequence-based scheme to avoid the challenges of chunk-wise processing lacks novelty.
2. How was the experiment in Figure 1c and Table 5 conducted?
3. Under the same token-level compression ratio, how long of a context can the proposed method process on a single machine, and how is its performance compared with other works?

---

> ### Author Rebuttal · Authors · 2025-07-30
>
> We are pleased to see your recognition of our work. Regarding the concerns you raised, we would like to offer a detailed clarification, which we believe will help you better understand the core contributions and novelty of our approach.
>
> > **Weakness 1:** Logical soundness and novelty of the method
>
> We understand the reviewer's emphasis on chunk-wise schemes for long-context scenarios. However, there may exist two potential key misunderstandings in this comment, which we wish to clarify here:
>
> 1. In the training stage, backpropagation still requires retaining the computational graph and intermediate activations even with the chunk-wise input. **Therefore, chunk-wise training cannot reduce peak GPU memory usage.** Furthermore, the chunk-wise forward typically leads to a complex computational graph and severe memory fragmentation. This reduces hardware utilization and hinders training speed. In contrast, UniGist's proposed one-pass full-sequence training, combined with the unified sparse gist layout, not only enhances context compression quality but also yields significant training efficiency advantages through better kernel design.
>
> 2. **UniGist does not mandate full-sequence processing during inference.** We fully agree that managing peak memory is a practical challenge during inference, and we wish to emphasize that **UniGist fully supports chunk-wise inference and in a more flexible manner.** Specifically: **(1) During the prefill phase:** Users can flexibly set the chunk size and apply sparse attention accordingly. As shown in Section 4.3 and Figure 4, UniGist's gist layout is inherently compatible with different chunk sizes without compromising compression quality. **(2) During the decoding phase:** UniGist offers a more efficient compression strategy than existing methods. Compression and kv cache eviction can occur after every r generated tokens (where r is the compression ratio, e.g., 4 or 8), rather than waiting for a full chunk (e.g., 2048 tokens) in previous gist methods. This mechanism reduces memory pressure during decoding and boosts throughput.
>
> Therefore, we wish to clarify that the core design principle of UniGist is to provide a more efficient and memory-friendly solution for both training and inference while simultaneously improving compression quality, rather than simply avoiding the challenges of chunk-wise processing. We believe this co-design of methodology and  optimizations offers a more fundamental and efficient solution than previous approaches, which constitutes the core contribution and novelty of our work.
>
> > **Weakness 2:** Experimental details in Figure 1c and Table 5
>
> Thank you for raising this point. We are happy to provide more detailed clarifications:
>
> - **Regarding Figure 1c:** As mentioned in Lines 127–133 of the paper, we evaluate a compression model trained using the traditional chunk-wise method. We randomly sampled 1k sequences of length 32K from the SlimPajama dataset. For each token position in a sequence, we computed the LM loss. To smooth the curve and highlight local trends, each data point in the figure represents the average loss over a sliding window covering the token itself and 16 neighboring tokens on both sides.
> - **Regarding Table 5:** The detailed hardware and model configurations have been provided in Appendix C under “Speed-up Details” (Lines 550–553). We used the standard triton.testing.do_bench function for accurate performance evaluation. This function performs multiple warm-up runs to eliminate initialization overhead, followed by several benchmark runs whose results are averaged to ensure stable and reliable latency measurements.
>
> We will include these implementation details in the final version of the appendix to further enhance the transparency and reproducibility of our work.
>
> > **Weakness 3:** Maximum Context Length and Performance on a Single GPU
>
> To further address your concerns regarding the long-context processing capability on a single GPU, we conducted additional experiments. We benchmarked the peak GPU memory usage of different strategies for processing extremely long contexts on a single 40GB GPU. All experiments were based on the Llama-3.1-8B-Instruct model. The results are presented below:
>
> | Method | Compression Ratio | 128K | 256K | 320K | 512K |
> | :---: | :---: | :---: | :---: | :---: | :---: |
> | UniGist | 4 | 24.3GB | 32.6GB | 36.7GB | OOM |
> | UniGist | 8 | 20.2GB | 26.4GB | 30.5GB | 37.7GB |
>
> **The results clearly demonstrate that UniGist's KV Cache compression technique significantly extends the maximum context length manageable on a single GPU.** Specifically, with a compression ratio of 4, UniGist can process contexts up to 320K on a single 40GB GPU (using 36.7GB of memory), suggesting it can support lengths approaching 340K. When the compression ratio is increased to 8, processing a 512K context requires only 37.7GB of memory, indicating that the supported context length can extend well beyond 512K. In contrast, a standard Full Attention model under the same hardware configuration can only handle a maximum context of approximately 80K. This comparison robustly demonstrates the practical value and substantial advantages of UniGist in extending the long-context processing capabilities of models.
>
> Thank you again for your valuable time and constructive feedback. We hope that our detailed responses and specific revision plan have clearly addressed your concerns. We appreciate your consideration and look forward to your feedback.

---

> > ### Comment · Reviewer_7dFr · 2025-08-03
> >
> > Thanks for the reply. It addressed my concerns. Based on other reviews and the rebuttal, especially the clarification of UniGist's flexibility and the supplement of supported maximum context length on a Single GPU, I will raise my rating to Borderline Accept.

---

> > > ### Author Response · Authors · 2025-08-04
> > >
> > > Thanks for your reply and for your willingness to consider raising the rating! We’re very glad to hear that our response has addressed your concerns. Your positive feedback is truly encouraging to us. We sincerely appreciate your time and effort throughout the review process.

---

### Official Review · Reviewer_upn7 · 2025-07-02

**Clarity:** 2
**Significance:** 2
**Originality:** 3
**Rating:** 4
**Confidence:** 3

**Summary:**

This paper proposes a new gist-based sequence-level compression method that efficient long context modeling without chunk-wise training. In order to improve throughput for the sparse attention, a gist shift trick (moving all gist tokens to the rightmost position of the sequence) is proposed. The experiments on several long-context tasks demonstrate the good performance of this proposed method.

**Questions:**

1. It is unclear why chunk-based training is used in the previous method. Is it to save memory? If it is, is your method more memory-consuming for other GIST-based method?

**Ethical Concerns:**

["NO or VERY MINOR ethics concerns only"]

**Final Justification:**

The rebuttal addresses my concerns on the presentation and contributions. So I keep this score as 4.

**Quality:**

3

**Strengths And Weaknesses:**

Strengths:

1. Some analysis on previous GIST architectures:  ignore compressed contexts and high loss on the start part of each chunk. This clearly show the issues of the previous GIST architectures.

2. A new gist-based sequence-level compression method is proposed. Some new techniques ( (attention sink prefix, position encoding alignment, local window) ) are used in the new method.

3. The proposed method shows the consistent improvements on several long-context tasks. Two different-size models (Llama3.1-8B-Instruct and Llama-3.2-3B-Instruct) comparison are also shown.


Weaknesses:

1. Some presentation are not clear. In Figure 1(b), there is no any annotation what are the attention weight range for each color. What are the supervised tuning on 1B tokens of mixed instruction data? What are the synthetic dataset? Otherwise, it is difficult to evaluate the generation ability of the proposed method.

2. In the proposed methods, there are two main added techniques (sink tokens and local window). In section 5.3 some ablation is done. However, ablation is only on training curves (loss, and gradient norm). There is no any study on the effect of the performance.  These can help to know how importance of each part.

3. It is not surprised that local window with raw tokens can help the performance. However, it could hurt the efficiency. In Figure 7, there is  a comparison of efficiency between UniGist and standard full attention. There is no comparison with previous GIST method.

---

> ### Author Rebuttal · Authors · 2025-07-30
>
> Thanks for your review and constructive feedback. We are very pleased that you recognize the motivation for our work and the strong performance of our method. Regarding the concerns you have raised, we appreciate this opportunity to offer more in-depth clarifications and additional results.
>
> > **Weakness1:** concerns about paper presentation
>
> - **On the color annotation of Figure 1(b):** Thank you for pointing this out. In the original figure, the color brightness represents the magnitude of the attention weights, where brighter colors indicate higher weights. This is consistent with the standard settings of Python's Matplotlib plotting. We will add a color bar and corresponding legend to Figure 1(b) in the final version to clearly and intuitively depict the distribution of attention.
>
> - **On the details of supervised fine-tuning and the synthetic dataset:** This might be a misunderstanding. In Appendix A.3 (Training Data Processing), Lines 534-537,  we have already mentioned that we primarily use the Magpie dataset and additionally integrated samples from tasks such as LongAlpaca and BookSum, refering to the approach of Activation Beacon. To fully address your concerns and eliminate any ambiguity regarding the composition of our training data, we will make two key additions to Appendix A.3 in the revised version: (1) We will explicitly list the specific number of samples and mixing ratios from each data source. (2) We will use more prominent formatting and a dedicated table to clearly present the data sources, sample counts, and primary task types.
>
> > **Weakness2:** concerns about the ablation study
>
> This is a very insightful comment, and we fully agree that reporting task-level performance is a more intuitive way to demonstrate the importance of each component. That said, we would like to clarify that the manuscript already includes ablation results and discussions related to key components. Our responses are as follows:
>
> - **Comparison with Activation Beacon as an ablation for gist attention**: As noted, the comparison with Activation Beacon can be interpreted as a holistic ablation. Beacon adopts a traditional chunk-wise training strategy, whereas our method introduces a unified attention mechanism. As shown in Table 1, UniGist significantly outperforms Beacon across multiple tasks. This highlights the superiority of our overall framework.
>
> - **Local window ablation is already included:** The effect of local window size is illustrated in Figure 6 (Right), which shows its direct impact on performance in the RULER task. To further improve clarity, we will augment this figure with additional performance data from other tasks.
>
> - **Training stability is a critical prerequisite for performance:** Training stability is essential for learning effective compression. As shown in the left and middle parts of Figure 6, removing sink tokens leads to frequent gradient spikes and unstable convergence, which directly harms downstream performance. To further address your concern, we conduct new experiments by continue pretraining a Llama-3.2-3B-Instruct model for 4B tokens and evaluate its long-context performance. We can observe that introducing sink tokens generally leads to better performance on the final tasks based on the following table. These discussions and new experimental results will be added to the appendix in the revised manuscript.
>
> | Sink size | RAG | Rerank | LongQA | ICL | Synthetic | Summ. |
> |:---|:---:|:---:|:---:|:---:|:---:|:---:|
> | sink=0 | 51.5 | 2.9 | 37.9 | 71.4 | 68.0 | 21.4 |
> | sink=128 | 53.3 | 3.7 | 38.4 | 72.0 | 72.1 | 21.4 |
>
> > **Weakness3:** concerns about efficiency (local window and comparison with previous gist models)
> We provide responses to both aspects of your concerns below:
>
> **Part A: The computational overhead introduced by the local window is negligible**
>
> - **Theoretical Analysis:** The local window size is typically set to a small value (e.g., 32, 64, 128) to achieve the desired performance gains. In our block-wise sparse attention kernel (with block size typically set to 128), this only introduces one additional key-value block computation at the per-query block level. Compared to the entire sequence length, especially in long-context scenarios, this overhead is almost negligible.
> - **Empirical Validation:** To address your concern more thoroughly, we conducted a new experiment measuring the attention forward and backward pass latency with and without the local window for a 64K-length sequence as shown in the following table. The results strongly demonstrate that the overhead introduced by the local window is negligible, whereas its performance benefits are substantial. This validates the superiority of our design.
>
> | Local window size | Forward (ms) | Backward (ms) |
> | :---: | :---: | :---: |
> | 0 | 22.1 | 141.5 |
> | 128 | 21.6 | 137.9 |
>
> **Part B: Compared to previous GIST methods, UniGist also holds a significant efficiency advantage during inference.**
>
> UniGist demonstrates significant inference efficiency gains over previous gist methods in two key phases:
>
> - **Faster Prefill Stage:** Our method allows for more flexible chunk-wise processing during inference. In previous gist methods, attention within each chunk was computed using a full attention. In contrast, UniGist leverages a custom sparse attention kernel to perform accelerated prefill by attending only to sink, gist, and local tokens within each chunk. This results in faster prefill speeds for the same chunk size.
> - **More efficient Decoding Stage:** Previous gist approaches typically wait until an entire chunk (e.g., 2048 tokens) is generated before performing compression. In contrast, UniGist can insert a new gist token and discard raw tokens every r tokens (where r is the compression ratio, e.g., 4 or 8), enabling fine-grained, more frequent compression. This leads to two major advantages: **(1) Lower peak KV cache usage:** We can release unused raw token cache earlier, maintaining lower memory usage throughout generation. **(2) Reduced computation:** Fewer KV are involved in attention computation, reducing both memory access and compute overhead for each next-token prediction.
>
> In summary, the local window introduces negligible overhead, and UniGist achieves superior inference efficiency through accelerated prefill and fine-grained decoding compression compared to prior gist methods. We will include this detailed analysis and new comparative experimental results in the final version of the Appendix.
>
> > **Question 1:** Why did previous methods use chunk-based training? Is it for memory savings? Is your method more memory-intensive than other GIST-based methods?
>
> We would like to clarify a key point: **most prior methods adopted chunk-wise training primarily for ease of implementation, not for memory savings.** They could reuse existing, highly-optimized kernels like FlashAttention for the full attention within each chunk, thus reducing implementation complexity.
>
> However, this design choice is a compromise, made at the cost of model performance, training efficiency, and inference flexibility. Furthermore, **chunk-wise training does not actually save memory during training.** The backward propagation requires retaining the computation graph and intermediate activations for all chunks, meaning the GPU memory occupied by a chunk cannot be freed immediately after it is processed.
>
> Specifically, chunk-wise training introduces the following limitations, which our work aims to address:
>
> - **Performance Limitations:** As illustrated in Figure 1(b), chunk boundaries bring unnecessary obstacles for learning compression. It leads to poor compressed context quality and imbalanced loss within chunks.
>
> - **Training Inefficiency:** Iteratively performing forward passes on the same Transformer block creates a complex computation graph and leads to significant memory fragmentation, which reduces throughput and hardware efficiency.
>
> - **Inference Inflexibility:** The chunk size used during inference is limited by the training configuration, making it difficult to adapt to different needs.
>
> **Under identical settings, UniGist is more memory-efficient and faster in both training and inference compared to previous gist methods, as detailed in our response to Weakness 3, Part B.** In short, we believe UniGist not only delivers significant improvements in model performance (as shown in Table 1) but also offers a more efficient and flexible compression to the previous chunk-wise approach.
>
> Thank you again for your valuable time and constructive feedback. We hope that our detailed responses and specific revision plan have clearly addressed your concerns. We appreciate your consideration and look forward to your feedback.

---

> > ### Comment · Reviewer_upn7 · 2025-08-05
> >
> > Thanks for the response from the authors. It addresses my concerns.

---

### Official Review · Reviewer_ZixF · 2025-07-03

**Clarity:** 3
**Significance:** 2
**Originality:** 3
**Rating:** 4
**Confidence:** 3

**Summary:**

This paper introduces UniGist, a sequence-level long context compression framework for large language models. The proposed approach moves away from the prevalent chunk-wise gist training strategies, instead enabling a chunk-free, unified attention pattern with learnable gist tokens interleaved across the entire sequence. The authors further introduce a “gist shift” trick—an efficient hardware-aligned kernel design—to exploit GPU architecture, accelerating both training and inference. Experiments on long-context tasks demonstrate that UniGist significantly outperforms previous gist-based compression methods.

**Questions:**

Please refer to the weaknesses.

**Ethical Concerns:**

["NO or VERY MINOR ethics concerns only"]

**Final Justification:**

In the rebuttal, the authors addressed the core differences between Unigist and previous works, and explained why this approach achieves more effective compression. Therefore, I have raised my score.

**Limitations:**

Yes

**Quality:**

3

**Strengths And Weaknesses:**

**Strengths**
* Clear motivation: The limitations of prior chunk-wise gist-based approaches are clearly identified and experimentally analyzed.
* Performance:Empirical evaluations demonstrate that UniGist approaches full attention performance on long-context tasks, particularly outperforming other compression methods by a large margin. Also, the gist shift trick is an engineering contribution that aligns the sparse attention pattern with GPU-friendly block processing.

**Weaknesses**
* Block-wise sparse attention has been studied in numerous previous works [1], and various variants of attention patterns have been proposed recently. It would be better to clarify the unique characteristics of the attention pattern in this paper to better validate the novelty of the work.
* The main baselines for Unigist are approaches that adopt a chunk-wise strategy. Although they result in a loss of contextual information, their primary advantages lie in acceleration and memory savings. However, the paper does not compare with them in these two aspects.

[1] Generating Long Sequences with Sparse Transformers

---

> ### Author Rebuttal · Authors · 2025-07-30
>
> Thanks for your insightful feedback. We are pleased that you recognized the motivation and empirical strength of our work. In response to the points you raised, we would like to take this opportunity to offer further clarification:
>
> > **Weakness 1:** Block-wise sparse attention has been studied in numerous previous works [...] it would be better to clarify the unique characteristics of the attention pattern in this paper to better validate the novelty of the work.
>
> We fully understand the concern: given the extensive prior research on sparse attention, does the attention pattern used in UniGist constitute a genuine innovation? We'd like to clarify that our key innovation lies not in the sparse pattern itself, but rather to enable **effctive and efficient context compression**, which fundamentally differentiates our approach from conventional sparse attention methods.
>
> 1. **UniGist’s goal differs fundamentally from prior sparse attention works** (e.g., Sparse Transformer, Longformer, BigBird). Those methods aim to approximate full attention more efficiently, but still require caching all original tokens' kv cache during inference, thus failing to truly alleviate memory pressure in long-context scenarios. In contrast, UniGist explicitly discards original tokens after compressing them into a small number of learnable gist tokens, enabling sequence-level compression that substantially reduces KV cache size during inference.
> 2. **Moreover, the attention pattern in UniGist is uniquely constructed around connections to the gist tokens, rather than being fixed to blocks or sliding windows.** This offers flexibility to support context compression with adjustable chunk sizes as demonstrated in Section 4.3 and Figure 4. This is not feasible with traditional sparse attention designs.
>
> Therefore, the novelty of our work lies not in the geometric shape of the sparse pattern itself,  but in its effective and efficient compression characteristics. We will make this fundamental distinction clearer in the revised version of the paper to address your concern.
>
> > **Weakness 2:** The main baselines for Unigist are approaches that adopt a chunk-wise strategy. Although they result in a loss of contextual information, their primary advantages lie in acceleration and memory savings. However, the paper does not compare with them in these two aspects.
>
> We appreciate your insightful observation regarding the efficiency strengths of chunk-wise approaches. However, we would like to clarify a potential misunderstanding: **UniGist is fully compatible with chunk-wise compression during inference, and achieves comparable or even better acceleration and memory efficiency.**
>
> To comprehensively address this concern, we believe it is important to distinguish between the **training and inference stages**, as UniGist introduces fundamental improvements in both.
>
> 1. **Training Stage:** UniGist achieves better compression quality and training efficiency. A common misconception is that chunk-wise strategies also save memory or accelerate training. In practice, this is often not the case. Backward propagation requires the computation graph and intermediate activations for the full sequence, meaning chunk-wise training can not reduce peak memory usage. On the contrary, iterative forward passes on the same module usually create complex computation graphs and severe memory fragmentation, which degrades hardware utilization and lowers training throughput. UniGist addresses these issues through its one-pass training paradigm and our custom hardware-aligned kernel, achieving higher compression quality alongside faster training speeds.
>
> 2. **Inference Stage:** UniGist not only supports chunk-wise processing but also surpasses prior methods in both flexibility and efficiency. **(1) UniGist is fully compatible with a chunk-wise strategy during the prefilling stage to manage peak memory.** This ensures that for the same chunk size, UniGist's peak memory usage is comparable to that of previous chunk-wise methods. More importantly, thanks to our decoupled design, the chunk size can be flexibly configured at inference time without any need for retraining. **(2) Prior methods typically perform full attention within each chunk.** UniGist, even during chunk-wise prefilling, can leverage our designed sparse attention kernel for acceleration, making it more efficient than baselines even in this step. **(3) Previous methods must accumulate to a full chunk (e.g., 2048 tokens) before performing a single compression step.** In contrast, UniGist can perform compression and discard the corresponding KV cache immediately after every r new tokens are generated (where r is the compression ratio). This fine-grained compression mechanism reduces peak memory, computation, and memory access during the decoding phase.
>
> **To fully address your concern, we have conducted a new end-to-end efficiency experiment.** We benchmark UniGist against two representative baseline methods using the Llama-3.1-8B Instruct model in a 64K context (with a chunk size of 2K). As shown in the table below, UniGist achieves the best performance on key metrics such as TTFT, clearly demonstrating its  superior overall efficiency:
>
> | Method | Compression Ratio | TTFT (ms) | TPOT (ms/token) | End-to-End Latency (ms) |
> | :---: | :---: | :---: | :---: | :---: |
> | Full Attention | - | 9484.7 | 50.5 | 15900.9 |
> | Beacon | 4 | 8749.6 | 37.9 | 12561.5 |
> | UniGist | 4 | 8587.4 | 37.0 | 13284.0 |
>
> We will strengthen this analysis in the revised manuscript and include the new quantitative comparison in the appendix to provide clearer evidence of UniGist’s efficiency advantages.
>
> Thank you again for your valuable time and constructive feedback. We hope our detailed responses and the planned revisions have thoroughly addressed your concerns. We appreciate your consideration and look forward to your feedback.

---

> > ### Comment · Reviewer_ZixF · 2025-08-06
> >
> > Thank you for your detailed reply, which has helped me gain a clearer understanding. Here, I have a further question: the paper mentions that a core challenge is the issue of cross-chunk information flow. Could you please more clearly explain the core differences between Unigist and previous sequence-level compression works, such as activation beacon? As I understand it currently, the difference in design lies in the fact that Unigist uses sink tokens at the beginning of the entire sequence.

---

> > > ### Author Response · Authors · 2025-08-07
> > >
> > > Thank you for your follow-up question. It allows us to further clarify the core mechanism of UniGist. Our innovation is not simply about adding sink tokens. **Instead, it centers on the Unified Sparse Gist Layout and a corresponding decoupled training-inference strategy.** Below, we explain how our design fundamentally differs from methods like Activation Beacon.
> > >
> > > > **1. Key Mechanism: The Unified Sparse Gist Layout**
> > >
> > > Methods like Activation Beacon employ a "chunk-wise training" paradigm. In this setup, the model processes inputs in the form of `[historical gists] + [current full chunk]`. This design allows the model to easily rely on the large number of raw tokens within the current chunk to minimize its loss, creating a "learning shortcut" that circumvents the need to learn real context compression.
> > >
> > > To eliminate this shortcut, we propose the **Unified Sparse Gist Layout**. Our layout removes the reliance on a "current full chunk." In our design, each raw token $x_i$ has access only to a limited raw local context. Long-range context must be accessed through globally distributed gist tokens. **This forces the model to learn to encode and retrieve information using gist tokens, rather than only using raw tokens nearby.** This enables effective long-range information flow across the sequence.
> > >
> > > > **2. Decoupled Training and Inference Strategy**
> > >
> > > Our design separates the training and inference processes to optimize both learning and efficiency:
> > >
> > > - **Training Phase: Higher compression quality and better hardware utilization.** Chunk-wise training, despite its chunked input, **can not reduce peak GPU memory** during backpropagation, as the entire computation graph must be preserved. More importantly, the iterative forward passes in chunk-wise training often lead to complex computation graphs and severe memory fragmentation, which degrades hardware utilization and slows down training. In contrast, UniGist adopts one-pass full-sequence training and a hardware-friendly kernel to improve compression quality while significantly increasing training speed and hardware utilization.
> > >
> > > - **Inference Phase: More flexible and efficient chunking.** Although training is global, inference can still use chunk-wise compression to save memory. Since the model has learned the unified layout, it can effectively retrieve global information from gist tokens at any position. This provides two key benefits: **(1) Flexible Prefill:** Users can flexibly set the chunk size to adapt to different hardware or task requirements, while still applying sparse attention. As shown in Section 4.3 and Figure 4 of our paper, UniGist's layout naturally accommodates variable chunk sizes without being tied to the chunk size used during training like previous methods. **(2) Efficient Decoding:** UniGist enables finer-grained compression. It can perform compression and memory discard after generating only a few tokens (e.g., every 4 or 8 tokens), instead of waiting for a full large chunk (e.g., 2048 tokens) like previous methods. This reduces memory usage and increases decoding throughput.
> > >
> > > This **"global training, chunked inference"** strategy ensures the model effectively learns long-range dependencies, retains the memory-saving benefits of chunked inference, and introduces flexibility and efficiency. This is fundamentally different from prior approaches, which tightly couple both training and inference to fixed chunk sizes.
> > >
> > > We summarize the key differences in the table below:
> > >
> > > | Feature | Previous methods | UniGist (Ours) |
> > > |--------|----------------------------------|----------------|
> > > | **Training Paradigm** | Chunk-wise training leads to compression bottleneck | Global one-pass training. Learns in continuous global context and avoids shortcuts. |
> > > | **Training Efficiency** | Complex computation graph, memory fragmentation, low hardware utilization. | Unified computation graph with custom kernel. Higher hardware efficiency and training speed. |
> > > | **Inference Prefill Flexibility** | Strongly tied to training chunk size. Limited adaptability. | Decoupled from training. Chunk size is flexible across hardware and tasks. |
> > > | **Inference Decoding Efficiency** | Coarse compression: compresses only after full chunk (e.g., 2048 tokens). | Fine-grained compression: compresses every few tokens (e.g., 4 or 8). Higher throughput. |
> > >
> > > We hope this clarification can fully address your concern. If you have any further questions, we would be glad to provide additional explanation. We believe that UniGist offers a more general and effective solution for sequence-level compression, and we sincerely look forward to your positive reconsideration of our work. Thank you again!

---

> > > > ### Comment · Reviewer_ZixF · 2025-08-09
> > > >
> > > > Thank you for the authors' detailed reply, which has addressed my concern about the innovativeness of Unigist. Thus, I will raise my rating to Borderline Accept.

---

> > > ### Author Response · Authors · 2025-08-08
> > >
> > > Dear Reviewer ZixF,
> > >
> > > Thank you again for your insightful follow-up question. It provided us with a valuable opportunity to further clarify the core contributions of our work, UniGist. As the discussion period is drawing to a close in less than 24 hours, we are writing to you with a sincere request for your final consideration.
> > >
> > > We would like to gently share that the other reviewers have expressed positive assessments of our work. Your expert opinion is incredibly important to us. Your thorough reviews have been highly encouraging and instrumental in helping us strengthen our paper, and we truly hope to earn your support as well.
> > >
> > > We believe our last detailed response, particularly the clarification on the Unified Sparse Gist Layout and the decoupled training-inference strategy, summarized in the comparison table, can hopefully address your concerns regarding the core differences from prior works like Activation Beacon. We would be deeply grateful if you could take another look at our clarification.
> > >
> > > Of course, should any part of our explanation remain unclear, or if you have any further questions, please do not hesitate to let us know. We are on standby and will do our utmost to provide a prompt and satisfactory answer.
> > >
> > > Thank you once again for your invaluable time and dedication to reviewing our manuscript. We eagerly await your feedback.
> > >
> > > Best regards,
> > >
> > > The Authors

---

### Official Review · Reviewer_XQBu · 2025-07-03

**Clarity:** 3
**Significance:** 2
**Originality:** 2
**Rating:** 4
**Confidence:** 4

**Summary:**

This paper overcomes some of the limitations of efficient Transformers based on gist tokens. After finding that attention remains localised within chunks and that early tokens within chunks exhibit higher loss, the Authors propose a chunk-free variant of gist tokens. This is achieved through a special mask (while taking care of sink tokens and positional encoding). To this end, they design a special attention kernel efficiently supporting the required masking by restructuring the mask matrix. As a result, they report minor degradation in long-context performance, while peak memory and latency are substantially reduced.

**Questions:**

- How does position encoding alignment work with relative positions (e.g., RoPE)?
- How are gist tokens initialised in practice?

**Ethical Concerns:**

["NO or VERY MINOR ethics concerns only"]

**Final Justification:**

Based on the discussion on the choice of baselines and the new end-to-end performance evaluations, I maintain my positive assessment of this paper.

**Limitations:**

The main limitation, which is not properly discussed, is the overhead of (continued) training a model, which may be prohibitively expensive. In addition, no discussion of broader societal or environmental implications is present in the main text.

**Quality:**

2

**Strengths And Weaknesses:**

**Strengths**
- The analyses on the behaviour of current gist-based models and their limitations (section 3.2) are quite insightful.
- The improvements over gist-based models are meaningful and allow for better accessing gist tokens beyond the current chunk.
- The results

**Weaknesses**
- The choice of baselines is inadequate: training-free methods do not represent the state of the art in sparse attention / KV cache compression. Based on large-scale experiments in Sparse Frontier (Nawrot et al. 2025), these are rather MInference for pre-filling and Quest or cartridges (Eyuboglu et al. 2025) for decoding during inference. As for training-based baselines, again, state-of-the-art methods beyond gist-based ones are ignored, e.g., Deepseek’s NSA. To understand if the proposed method is competitive, it must be compared with additional baselines.
- Considering that one of the main purposes of KV cache compression is to speed up inference, I was a bit surprised by how little of the paper is dedicated to properly measuring this property. Lines 295-301 only report forward / backward latency in an attention layer, rather than the actual runtime on specific accelerators, or the latency-throughput relationship for different batch sizes and sequence lengths.

**Minor**
- The problem statement in the introduction could be improved: while “the cost of self-attention” is relevant for both training and inference, in the latter it matters mostly only during pre-filling, but not as much for decoding, as this phase is mostly memory-bound. This should be clarified a bit better (also to explain why reducing KV cache size is not only related to VRAM memory load but also to throughput). Overall, the introduction would benefit from identifying different phases, their efficiency bottlenecks, and how different strategies (including but not limited to gist-based models) may overcome each of them specifically.
- “Previous empirical study shows […]” (line 102): This statement seems to warrant some citations.
- For continued training, you may consider distillation from the vanilla model rather than cross-entropy with the data (Łańcucki et al. 2025, Dynamic Memory Sparsification), as Llama’s data mix is unavailable and hence Prolong may cause a distribution shift.
- Which mixed instruction data was used for continued training (line 210)? This detail is important for reproducibility.

---

> ### Author Rebuttal · Authors · 2025-07-30
>
> We are grateful for your thoughtful review and constructive suggestions. We are particularly encouraged that you recognize our strengths. We would like to take this opportunity to offer further clarification and address your concerns:
>
> > **Weakness 1:** The choice of baselines is inadequate.
>
> First, we would like to clarify the core difference between our method and sparse attention methods.
> - **Our method, UniGist, is a context compression technique.** Its central idea is to replace the KV cache of a large number of original tokens using learnable "gist" tokens. This means the model can no longer access the original tokens after compression, which fundamentally reduces memory requirements.
> - In contrast, methods like Minference or Deepseek's NSA, which you mentioned, are essentially **lossless sparse attention**. While they employ various strategies to limit the number of key tokens each query token can attend to, the complete KV cache is still retained in memory. In principle, any original token remains accessible to future tokens.
>
> Therefore, a direct performance comparison between these two types of methods would be **unfair and out of scope in some extent**. Sparse attention methods naturally have an advantage in detail-recalling tasks, whereas our method excels in memory efficiency.
>
> Despite this fundamental difference, we fully understand your desire to see how our work competes in a broader context. To address your concern and  demonstrate the comparative effectiveness of our approach, we conduct new experiments comparing UniGist with MInference as a representative (based on Llama3.1-8B-Instruct):
>
> | Method | RAG | Rerank | LongQA | ICL | Synthetic | Summ. |
> |:---|:---:|:---:|:---:|:---:|:---:|:---:|
> | Full Attention | 61.9 | 51.4 | 43.5 | 81.8 | 99.3 | 28.9 |
> | MInference | 60.8 | 50.9 | 46.1 | 36.6 | 95 | 28.5 |
> | UniGist | 59.5 | 46.7 | 44.7 | 83.4 | 98.3 | 26.7 |
>
> We plan to add these new experimental results and a detailed discussion to the Appendix. This will allow us to fully address your concern.
>
> > **Weakness 2:** Lines 295-301 only report forward / backward latency in an attention layer, rather than the actual runtime on specific accelerators, or the latency-throughput relationship.
>
> First, we would like to clarify our rationale for initially choosing to evaluate attention latency. As stated in lines 295-296 of our paper, our method's primary improvements are in attention efficiency and GPU memory usage. **Therefore, isolating the attention layer for evaluation allows for an objective and clear demonstration of the direct speed-up provided by UniGist, avoiding interference from other model components.** We believe this "controlled experiment" is crucial for validating the core contribution of our method.
>
> At the same time, **we would like to point out that in our submitted version, we have already provided the raw latency data (in ms) on actual hardware in Appendix C, Table 5**, which serves as a foundation for end-to-end performance analysis.
>
> To thoroughly address your concerns and provide a more comprehensive view  of UniGist's practical efficiency advantages, we follow your valuable suggestion and conduct additional **end-to-end performance evaluations on the Llama-3.1-8B-Instruct model and 64K context (output 128 tokens)**. Below is an example of the supplementary results:
>
> | Method | Compression Ratio | TTFT (ms) | TPOT (ms/token) | End-to-End Latency (ms) |
> | :---: | :---: | :---: | :---: | :---: |
> | Full Attention | - | 9484.7 | 50.5 | 15900.9 |
> | UniGist | 4 | 8587.4 | 37.0 | 13284.0 |
> | UniGist | 8 | 6581.6 | 33.2 | 10795.2 |
>
> We plan to add these experiments and discussion to Appendix C  in the revised manuscript. This will provide a more complete performance profile for our method.
>
> > **Minor1:** Overall, the introduction would benefit from identifying different phases, their efficiency bottlenecks, and how different strategies (including but not limited to gist-based models) may overcome each of them specifically.
>
> Thank's for your suggestion. We agree that a clearer distinction between the efficiency bottlenecks would significantly strengthen the introduction. In the revised manuscript, we will refine the second paragraph to address this explicitly. The new discussion will differentiate between the compute-bound pre-filling phase and the memory-bandwidth-bound decoding phase. Furthermore, it will clarify that the benefit of reducing the KV cache extends beyond simply saving memory; critically, it enhances decoding throughput by minimizing the data transferred from HBM to SRAM  in each step. This enhancement will make the motivation for our work more precise.
>
> > **Minor2:** “Previous empirical study shows […]” (line 102): This statement seems to warrant some citations.
>
> Thank you for pointing this out. The statement is primarily based on the findings from works such as **Activation Beacon [54] and the Gist Study [11]**,  which verified that this interleaved gist token insertion is an effective baseline. We will add these citations to the revised manuscript.
>
> > **Minor3:** On the suggestion of using distillation for continued training
>
> This is a very insightful suggestion! In our current work, we adopt the standard continued pre-training approach to maintain consistency with the training paradigm of related works, such as Activation Beacon. We will add a discussion of this strategy in the Appendix or in the future work section. We will acknowledge that distillation could be a more robust training method and frame it as a promising direction for future research.
>
> > **Minor4:** On the lack of detail for the mixed instruction data
>
> This appears to be a misunderstanding. In the current version of the paper,  we mention our supervised fine-tuning data composition in **Appendix A.3 (Training Data Processing, Lines 534-537)**, noting that it largely follows the setup of Activation Beacon and providing the corresponding citation. To address your concern and eliminate any ambiguity, we will make two key additions to Appendix A.3  in the revised manuscript: 1. We will explicitly list the specific sample counts and mixing ratios from each data source (e.g., Magpie,  LongAlpaca, and BookSum). 2. We will use a dedicated table to clearly present the data sources, sample counts, and primary task types.
>
> > **Question 1:** Regarding the compatibility of position encoding alignment with RoPE
>
> **Our position encoding alignment strategy is fully compatible with RoPE.** The core of RoPE is to apply a rotation matrix based on a token's absolute position index. Our strategy (as shown in Figure 2c) adjusts these position indices before they are used to compute the RoPE embeddings. Specifically, for an original token $x_i$, we still use its original position index $i$. The relative position difference between any two original tokens $x_i$ and $x_j$ remains $j-i$,  perfectly preserving their relative relationship. The gist tokens simply "borrow" the position of a subsequent token without disrupting the structure of the original token sequence. We will clarify this mechanism more explicitly in Section 4.1.
>
> > **Question 2:** Regarding the initialization of Gist Tokens
>
> Thank's for this question about the implementation detail. Gist tokens are learnable virtual tokens. In practice, we randomly initialize their embeddings, typically from a standard normal distribution, which is similar to the initialization of other special tokens like `[EOS]`. This embedding is then learned and updated end-to-end along with the other model parameters during training. We will add discussion about this detail to Section 4.1.
>
> Thank you again for your valuable time and constructive feedback. We hope that our detailed responses and specific revision plan have clearly addressed your concerns. We appreciate your consideration and look forward to your feedback.

---

> > ### Comment · Reviewer_XQBu · 2025-08-06
> >
> > Thank you for your response to my review.
> >
> > **Weakness 1: choice of baselines.**
> > While I agree that the scope identified by the Authors for their work does not encompass methods beyond gist tokens, the claims regard long-context modelling abilities more broadly. Hence, all competitive methods enabling long-context modelling are suitable baselines, including sparse attention. If the focus is instead purely on reducing memory load via KV cache compression, the paper should reflect this distinction better. In addition, some of the baselines I mentioned are still applicable, such as Cartridges. Having said that, I appreciate that the Authors reported additional results with a MInference baseline.
> >
> > **Weakness 2: end-to-end performance evaluations**
> > Thank you for sharing your new results for end-to-end performance evaluations. These show that higher compression ratios actually translate into lower TTFT, TPOT, and latency. Hence, I consider my concern on this point fully resolved.
> >
> > Overall, I maintain my positive assessment of this paper.

---

### Comment · Area_Chair_xAfk · 2025-08-05
**Please read the authors’ rebuttal and join the discussion**

Dear Reviewers,

Thank you for your valuable contributions.

The authors have provided rebuttal to your reviews. Please carefully read their responses  as soon as possible, and indicate whether your concerns have been addressed. You are also encouraged to engage in discussion with fellow reviewers.

Note that simply **submitting "Mandatory Acknowledgement" without posting any feedback is NOT allowed**. *Let's be the kind of reviewers we’d appreciate for our own submissions*.

Best,

AC

---

### Decision · Program_Chairs · 2025-09-17

**Decision:**

Accept (poster)

**Comment:**

This paper introduces UniGist, a sequence-level long context compression framework designed to be both general across tasks and hardware-aligned for efficient deployment. Reviewers agreed that the problem addressed is important and timely, and they appreciated the technical idea of combining general compression with hardware efficiency considerations. Strengths noted include clear motivation, practical relevance, and evidence that the method offers competitive performance with meaningful efficiency gains. At the same time, reviews were cautious, with concerns about limited novelty relative to prior compression methods, and questions on whether the empirical evaluation fully support broad claims. The rebuttal helped clarify design choices and provided additional results, partially alleviating concerns. Given that the reviewers’ reservations were mostly about scope rather than fundamental flaws, and considering the strengths in problem importance, methodological soundness, and demonstrated potential, I recommend acceptance.